# Determination of Rhodamine 6G with direct immersion single-drop microextraction combined with an optical probe

**Arina Skok[1,2], Andriy Vishnikin[1]\*, Yaroslav Bazel[2], Ján Toth[2]**

**1** Department of Analytical Chemistry, Faculty of Chemistry, Oles Honchar Dnipro National University, Dnipro, Ukraine, **2** Department of Analytical Chemistry, Institute of Chemistry, Faculty of Science, Pavol Jozef Šafárik University in Košice, Košice, Slovak Republic

\* vishnikin@hotmail.com

**Data Availability Statement:** All relevant data are within the manuscript.

**Funding:** The author(s) received no specific funding for this work.

## Abstract

The combination of an optical probe and single-drop direct immersion microextraction (DI-SDME-OP) was used for the preconcentration and subsequent spectrophotometric determination of rhodamine 6G (Rh6G). The developed method is based on the formation of an ionic associate between Rh6G and picric acid at pH 3.0 and its extraction with amyl acetate. A microdrop of the organic phase was stably placed in the hole of an optical probe immersed in the sample solution. The absorbance of the extraction phase was monitored at 534 nm. The proposed method combines in a single step several stages of the analytical procedure, such as pre-concentration, phase separation, transfer of the extraction phase to the instrument and online measurement. The sensitivity of the proposed approach is not inferior to existing microextraction methods involving the combination of liquid-phase or solid-phase extraction with spectrophotometry or HPLC with a UV-Vis detector. The evaluation of the greenness of the developed method carried out by the AGREE method (0.58 points) showed that it outperforms other similar existing techniques using this parameter. The calibration plot for the determination of Rh6G by the DI-SDME-OP method was linear over the range of 10–500 nM with a correlation coefficient of 0.9956. The limit of detection was 3.4 nM. The accuracy and applicability of the method were evaluated by the determination of Rh6G in natural waters and lipstick.

## 1. Introduction

Contamination of the aquatic environment is an issue of growing concern. Synthetic organic dyes are underestimated as a source of environmental pollution [1]. Due to inefficiencies in the dyeing process, the textile industry loses up to 200,000 tons of organic dyes to wastewater each year [2]. The result is the generation and release of large amounts of wastewater into the environment. Toxic dyes are poorly biodegradable and in addition the strong color of the dyes creates serious aesthetic and pollution problems in wastewater disposal as they are visible even at very low concentrations [3]. Strong absorption of sunlight by dyes reduces the photosynthetic activity of aquatic plants, seriously threatening entire ecosystems [4].

**Competing interests:** The authors have declared that no competing interests exist.

**Fig 1. Molecular structure of Rhodamine 6G.**

Rhodamine 6G (Rh6G) is a highly water soluble derivative of the xanthene dyes, with the chemical structure shown in Fig 1. It has been used as a colorant in textiles and foods [5]. Due to the carcinogenic and mutagenic properties of rhodamine dyes, the US Food and Drug Administration has strictly prohibited their use in the food, pharmaceutical and cosmetic industries [6, 7]. Water that contains rhodamine dyes is a cause of irritation to the skin, eyes and respiratory system of humans. It is also medically proven that drinking water contaminated with rhodamine dyes is highly carcinogenic and toxic [8]. Therefore, it is important to monitor the concentration of rhodamine dyes in water and to detect its illegal use in highly consumed products, such as soft drinks and cosmetics.

Various methods, such as UV-Vis spectrometry [9–11] and HPLC [7, 12–14], have been applied for the determination of Rh6G. UV-VIS spectrometry is an important instrumental technique in this field due to its simplicity and lower cost compared to other methods. The two main limitations in the spectrophotometric determination of rhodamine dyes are the low analyte concentrations and the positive or negative influence of accompanying ions on the analyte signal [15]. Accordingly, to overcome these limitations, a preconcentration process should be used as an important step before instrumental analysis.

Separation/preconcentration methods, including solid-phase extraction, cloud point extraction, solvent extraction, ion-exchange extraction, etc., are widely used to solve these problems. The sensitivity of the different methods for the determination of rhodamine dyes with preliminary solid-phase and liquid-phase separation was similar at the concentration level of 1–3 μg L$^{-1}$ (2–6 nM) [16]. However, when analyzing large sample volumes, solid phase extraction suffers from analyte breakthrough.

The use of fluorimetric detectors greatly increases the sensitivity for the determination of xanthene dyes. Flow injection fluorimetry was used for Rhodamine B determination with a LOD of 0.5 nM [17]. Recently, the combination of an optical probe and direct immersion single-drop microextraction (DI-SDME) with fluorescence detection was proposed by our research team for preconcentrating and assaying Rh6G [18]. The method has a high sensitivity of 0.15 nM. However, the use of conventional light sources, such as a halogen lamp, resulted in moderate sensitivity, while the use of a laser or LED required a significant increase in the complexity of the measuring system.

Sample preparation is one of the most important steps when analyzing organic dyes using UV-Vis spectrometry. The application of solid-phase or liquid-phase extraction has serious

drawbacks such as significant losses and use of chemical solvents, large effluent volumes, low pre-concentration ratio and the need for multiple steps. Despite the widespread use of Rh6G, only two types of microextraction approaches–dispersive liquid–liquid microextraction (DLLME) [7, 9, 10] and hollow-fiber liquid-phase microextraction (HF-LPME) [19]–have been employed to extract traces of Rh6G.

Single-drop microextraction has great potential for achieving better sensitivity compared to dispersion-based approaches because of the lower volume of extraction phase and consequently higher enrichment factor that can be achieved. A common approach is to place the extraction phase at the tip of the microsyringe needle, which leads to problems with microdrop stability and severely limits the volume of the extraction phase to 2–3 μL. For compatibility with accessories commonly used in spectrophotometric measurements, the volume of the extraction phase should usually be considerably larger. This problem was partially solved by using an optical immersion probe both as a measuring device and to hold the extraction phase in the headspace [20] and direct immersion mode [21]. This approach allowed us to successfully combine into a single step the operations of preconcentrating and separating, transferring the analyte to the spectrophotometer and measuring and to perform the measurements online. Unfortunately, in the direct immersion mode, only a few organic solvents can be stably retained in the optical probe hole, and the stirring speed must be limited. In headspace mode, this problem becomes even more acute, and only aqueous solutions have a sufficiently high adhesion to the optical probe material. One possible solution is to place an optical probe with the extraction phase in a vessel located in the headspace above the solution to be analyzed [22, 23]. A significant complication in this case is that mass transfer in the extraction phase is greatly retarded due to the reduced surface area between the headspace and the acceptor phase.

The combination of the optical probe with DI-SDME-OP is promising for the development of simple, environmentally friendly, highly sensitive online procedures for the determination of various analytes. In the present paper, this method, as continuation of the previous work [18], was first applied for the preconcentration and direct spectrophotometric determination of trace amounts of organic dyes. The ionic associate of Rh6G with picrate anion was extracted with 55 μL of amyl acetate placed as a microdrop in the hole of the optical probe. The absorbance of the extraction phase was recorded online at 534 nm. The method was applied for the determination of Rh6G in natural waters and lipstick.

## 2. Materials and methods

### 2.1 Reagents and equipment

All chemicals used were of analytical grade purity. A stock solution of 1 M $Na_2SO_4$ was prepared by dissolving an appropriate amount of the salt in distilled water. An ammonium acetate buffer solution was prepared by mixing 1 M solutions of acetic acid and ammonium hydroxide. A preparation of picric acid moistened with water, $\geq$98% (Sigma-Aldrich), was used to prepare a 1 mM stock solution of picric acid by dissolving 57 mg of the compound in distilled water and diluting to the mark with distilled water in a 250-mL volumetric flask. A 1 mM stock solution of the chloride salt of rhodamine 6G was prepared by dissolving 48 mg of it in 5 mL of ethanol and diluting to the mark with distilled water in a 100 mL volumetric flask. Potential interferents at concentrations of 0.1 M were prepared by dissolving appropriate amounts of them in distilled water. The sample of red lipstick (Oriflame) was purchased from a local market.

A 1 cm double-pass optical probe (Expedeon, UK) connected to a USB 4000 fiber optic spectrometer (Ocean Optics, USA) and an LS-1 tungsten halogen light source (Ocean Optics, USA) was used as a microdrop holder and for spectrophotometric measurements. Data were

recorded using OceanView spectroscopy software. A magnetic stirrer with an RH digital heating model (IKA$^{®}$-Werke GmbH & Co. KG, Germany) was used to stir the solutions. A portable pH 70 Vio meter (XS instruments, Italy) was used for pH measurements.

## 2.2 Rh6G determination with DI-SDME-OP in water samples

The selected river water samples were pre-filtered through a glass fiber filter before analysis. A magnetic stirring bar was placed in a 15 mL glass vial, then 7.5 mL aliquot of sample solution, 1 mL 0.1 M $Na_2SO_4$, and 1 mL acetate buffer at pH 3.0 were added consecutively. The vial was held with a stand and clamp, and the optical probe was immersed into the solution. A 55 μL drop of amyl acetate was then gently fixed with a GC microsyringe in the hole of the optical probe, starting from its low inner part. The stirring speed was set to 800 rpm. Subsequently, 0.5 mL of 0.1 mM picric acid was added to the reaction mixture and the absorbance was immediately recorded with an optical probe using 534 nm as the analytical wavelength. The absorbance measured after 30 minutes of extraction was taken as the analytical signal. Between measurements, the optical probe was cleaned with ethanol and then water. Access to the field site was not required to collect natural water samples.

## 2.3 Rh6G determination in lipstick

A 50 mg sample of lipstick was accurately weighed and dissolved in 50 mL of distilled water at 60°C. After cooling, the undissolved particles were removed by filtration on a Schott filter and analyzed as described above.

## 3. Results and discussion

### 3.1 Optimization of extraction conditions. Selection of extraction solvent and counterion

At least two important requirements must be met when selecting the solvent for extraction in the developed method. First, it should have good analyte extraction capability, and second, it should be well retained in the optical probe hole. As shown in a previous study [21], most organic solvents cannot be firmly retained in the optical probe hole due to poor wettability of the optical probe material. For example, Rh6G can be extracted well with chloroform, even in the absence of a counterion, but this solvent does not hold well enough in the hole of the optical probe. Both toluene and amyl acetate are well retained by the optical probe. Amyl acetate was selected as the optimum solvent for extraction as it showed higher extraction efficiency than toluene and is more environmentally friendly.

The use of large anions as counterions to form ionic association complexes can significantly improve the extraction of Rh6G due to the formation of a neutral and more hydrophobic complex compound. Perchlorate, picrate and dodecyl sulfate were tested as counter ions. In the case of dodecyl sulfate, when the concentration was above the micellization point, the extractant droplet was no longer retained in the optical probe hole. At lower concentrations of dodecyl sulfate, an ionic associate was formed which was incompletely extracted. Rh6G was best extracted when picric acid was used to neutralize its positive charge. The extraction with perchlorate anion was also worse than with picric acid. Therefore, picric acid was chosen as a counterion to improve the extraction efficiency of Rh6G. The effect of picric acid concentration on the extraction efficiency of Rh6G was studied in the range from 0.5 μM to 30 μM for an Rh6G concentration of 0.1 μM (Fig 2A). In the range from 5 μM to 30 μM, complete dye binding was achieved, and the absorbance of the organic phase remained constant. A picric acid concentration of 5 μM was chosen as the optimal concentration.

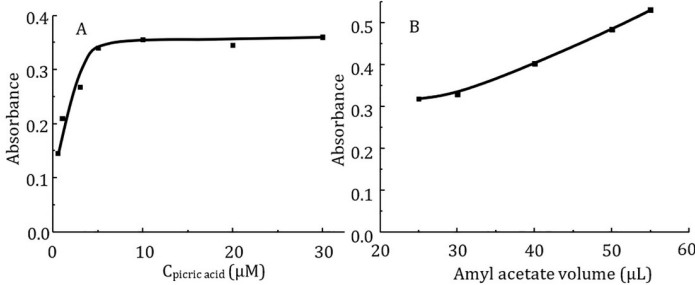

**Fig 2. Effect of picric acid concentration (A) and amyl acetate volume (B) on the absorbance of extractant phase.** Extraction conditions: Rh6G concentration 0.2 μM (A) or 0.25 μM (B); picric acid concentration 5 μM (A) or 30 μM (B); stirring speed 800 rpm; amyl acetate volume 55 μL; sodium sulfate concentration 0.1 M; extraction time 30 min; donor phase volume 10 mL; analytical wavelength 534 nm; cuvette path length 1 cm.

It was found that in the presence of salts such as sodium sulfate, the determination results become more reproducible. We believe that the salting out effect leads to a decrease in the solubility of amyl acetate in the aqueous phase, thus eliminating the effect of partial dissolution of the extractant in the aqueous solution. The absorbance of the organic phase stabilizes at salt concentrations between 2.5 mM and 0.1 M. A sodium sulfate concentration of 0.1 M was chosen as optimal.

### 3.2 Influence of extractant and donor phase volume

The choice of optimal extractant volume is limited by the probe hole size (Fig 3). A 20 μL drop of amyl acetate is not sufficient to completely cover the optical path of the probe, while drops of 60 μL or more are no longer retained in the probe hole. Accordingly, the effect of amyl acetate volume on the organic phase absorbance was investigated in the range of 25 to 55 μL. The absorbance gradually increased with increasing drop volume. A volume of 55 μl of amyl acetate was thus chosen as the optimal volume (Fig 2B). This dependence can be

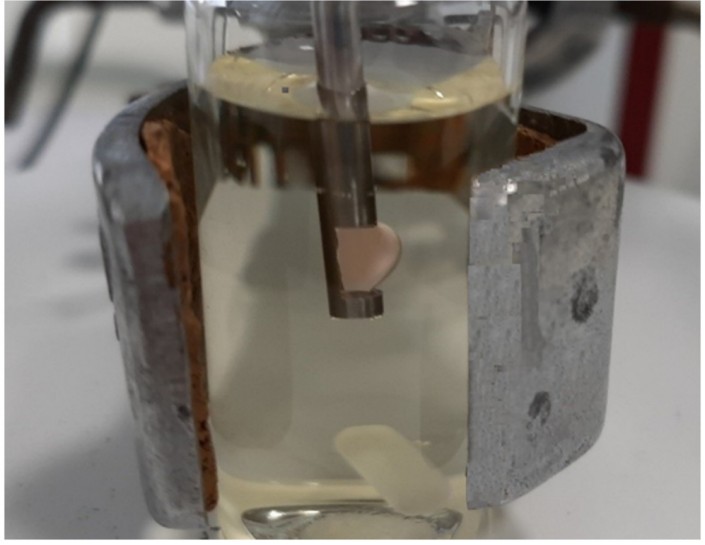

**Fig 3. View of the extraction system used for Rh6G microextraction after the extraction was completed.**

explained by considering the importance of the mass transfer rate through the surface area between the extraction solvent and the sample solution. Under the conditions investigated, extraction equilibrium cannot be reached in an acceptable time. Consequently, the extraction efficiency is not determined by the volume ratio of the sample and organic phase, but by the mass transfer rate of the analyte across the phase boundary. In accordance with this, an increase in surface area contributes to an increase in the extraction rate. A study of the effect of sample volume on the organic phase absorbance showed that increasing the volume of the donor phase from 10 to 20 mL had virtually no effect on the magnitude of the analytical signal. Again, the analytical signal was not improved by slowing down the mass transfer rate with a larger volume of aqueous phase. In further experiments, 10 mL of the total donor phase volume was used.

### 3.3 Effect of pH on the extraction

The p$K_a$ values for Rh6G and RhB are 6.13 and 3.22, respectively [5, 24]. They correspond to different protonation processes. For Rh6G, one of the two secondary amino groups is protonated, while for Rhodamine B the carboxyl group dissociates at pH $> \approx 3$. Both the molecular and protonated forms of Rh6G are well extracted by amyl acetate. However, as Fig 4 shows, the ionic associate formed between the cationic form of Rh6G and the picrate anion is extracted much better than the molecular form of this dye. The extraction behavior of RhB is distinctly different from that of Rh6G. In the presence of picrate anion, its protonated form is extracted quite well with amyl acetate. The zwitterionic form of RhB completely loses its extraction ability at pH $> 6$. A possible reason for this may be the hydration process of the negatively charged deprotonated carboxyl group. An ammonium acetate buffer with pH 3.0 was chosen as the medium for the determination of Rh6G. At this pH, co-extraction of Rh6G and RhB can occur. As shown in Fig 4, Rh6G extraction can be selective at pH $> 6$. In this case, it can be used for the selective determination of Rh6G in the presence of RhB or for their simultaneous determination.

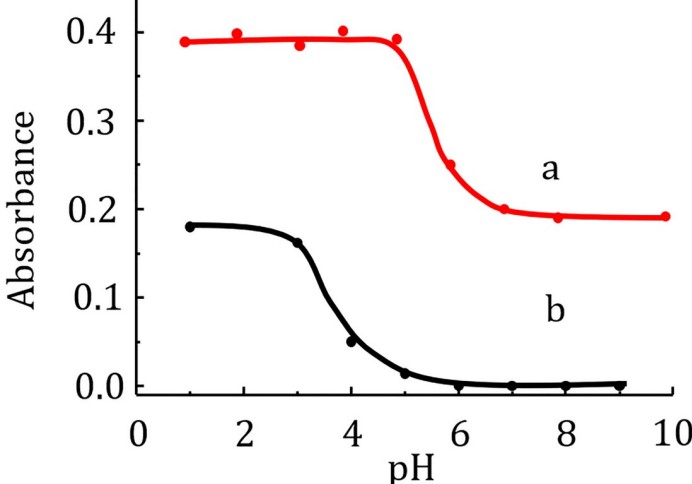

**Fig 4. Influence of donor phase pH on the extraction of Rh6G (a) and RhB (b).** Extraction conditions: Rh6G concentration 0.2 μM; RhB concentration 0.28 μM; picric acid concentration 5 μM; stirring speed 800 rpm; amyl acetate volume 55 μL; sodium sulfate concentration 0.1 M; extraction time 30 min; donor phase volume 10 mL; analytical wavelength 534 nm (a) or 555 nm (RhB); cuvette path length 1 cm.

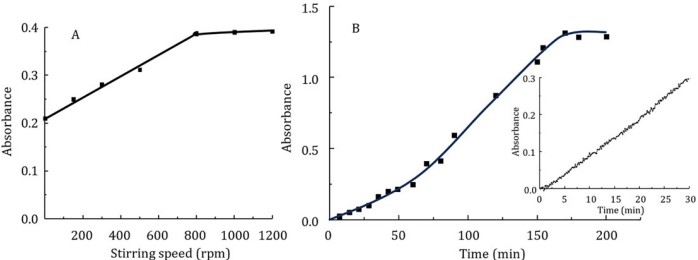

**Fig 5. Dependence of extractant phase absorbance on stirring speed (A) and extraction time (B) for extraction of 0.2 μM Rh6G with 55 μL amyl acetate.** Other conditions: picric acid concentration 5 μM, ammonium acetate buffer with pH 3.0, $Na_2SO_4$ concentration 0.1 M, donor phase volume 10 mL, extraction time 30 min (A), stirring speed 800 rpm (B).

## 3.4 Effect of stirring speed and extraction time

Stirring of the sample phase is necessary to increase mass transfer of the dye. However, a high stirring speed may lead to detachment of the extraction solvent from the optical probe. It was found that by using a smaller magnetic stirring bar (8.32 mm length and 3.01 mm diameter), the maximum stirring speed could be significantly increased, in this case to a maximum speed of 1000 rpm. Maximum organic phase absorbance was obtained at a stirring speed of 800 rpm (Fig 5A). Therefore, this stirring speed was chosen as optimal.

The extraction time has a great influence on the extraction efficiency. Two factors have a significant impact on the deterioration of extraction speed and efficiency in single-drop micro-extraction compared to classical extraction. The volume ratio of the sample and extraction phases increases significantly. At the same time, the surface area between the two phases becomes very small. Both of these obstacles slow down the mass transfer of the analyte. The role of mixing of the acceptor or donor phases increases. At a high ratio of donor and acceptor phases and very low concentrations the time to reach equilibrium increases sharply. In the studied concentration range for Rh6G, complete extraction takes more than 3 hours (Fig 5B). The optical probe allows obtaining reproducible kinetic data. With this in mind, the use of a fixed-time kinetic method allowed us to obtain well reproducible results for any extraction time. An extraction time of 30 min was chosen as a compromise between sensitivity and throughput of the assay (Fig 5B).

## 3.5 Interference study

Inorganic ions commonly present in environmental waters do not interfere with Rh6G determination (Table 1). Anionic and nonionic surfactants, represented by SDS and Triton X-114, have little effect on the determination of Rh6G. However, cationic surfactants can provoke the parallel extraction of picric acid. For example, interference from cetylpyridinium chloride is observed already at a concentration of 1 μM. Another source of interference is the co-extraction of other dyes. At pH 6.0, up to 0.5 mM RhB is tolerated, whereas fluorescein and Eriochrome Black T begin to interfere at 1 μM and 0.2 μM, respectively (Table 1).

## 3.6 Analytical characteristics of the DI-SDME-OP spectrophotometric determination of Rh6G

The calibration curve for the determination of Rh6G at pH 3 was linear over the range from 10 to 500 nM, with a correlation coefficient of 0.9956. The LOD and LOQ were calculated as the

**Table 1. Tolerance limits of potentially interfering ions and substances in the determination of 0.1 μM Rh6G.**

| Interferent | Tolerable interferent concentration (mM) |
|---|---|
| $F^-$, $Cl^-$, $Br^-$, $I^-$, $H_2PO_4^-$, $NO_2^-$, $NO_3^-$, $SCN^-$, $CO_3^{2-}$, | 10 |
| $SO_3^{2-}$ | 5 |
| $Na^+$, $K^+$, $NH_4^+$, $Ca^{2+}$, $Mg^{2+}$, $Zn^{2+}$, $Mn^{2+}$, $Ni^{2+}$, $Al^{3+}$, $Fe^{3+}$ | 10 |
| $Co^{2+}$, $Ba^{2+}$, $Cu^{2+}$ | 1 |
| $Cr^{6+}$ | 5 |
| SDS | 0.1 |
| Cetylpyridinium chloride | 0.001 |
| Triton X-114 | 10 |
| Fluorescein | 0.001 |
| Eriochrome Black T | 0.0002 |
| Rhodamine B | 0.5 [a] |

[a]At pH 6.0.

concentration equivalent to three-times and ten-times, respectively, the ratio of the standard deviation of the blank solution and the slope of the calibration plot. The LOD was equal to 3.4 nM (1.6 μg $L^{-1}$), and the LOQ was 10 nM (5.0 μg $L^{-1}$). Intraday RSD values varied from 1.7 to 3% for Rh6G concentrations from 50 to 150 nM. Interday RSD values increased up to 6% for 150 nM Rh6G concentration.

## 3.7 Analysis of real samples

The developed method was applied to the determination of spiked amounts of Rh6G in tap and river water, lipstick and model samples. The results are presented in Table 2. Dyes present in lipstick did not interfere with the determination of Rh6G. All analyses, except for the model sample, were performed at pH 3.

## 3.8 Comparison with other methods

The sensitivity of the proposed method is similar to existing spectrophotometric methods combined with liquid-phase microextraction methods [9, 10] (Table 3). However, DLLME-based methods require the use of larger volumes of organic solvents, including chloroform, which is toxic and harmful to the environment [9, 10]. In addition, the relatively low volume

**Table 2. Determination of Rh6G in water samples by DI-SDME-OP.**

| Sample | Added Rh6G (nM) | Found Rh6G (nM [b]) | Recovery (%) |
|---|---|---|---|
| **Tap water** | 49 | 50.0 ± 9.9 | 102.0 |
| | 86 | 82.1 ± 6.9 | 95.4 |
| **River water** | 49 | 52.3 ± 3.6 | 106.1 |
| | 98 | 106 ± 16 | 108.1 |
| **Model sample [a]** | 40 | 42.0 ± 2.7 | 105.0 |
| | 80 | 81.8 ± 5.0 | 102.3 |
| **Lipstick** | 40 | 42.6 ± 5.3 | 106.5 |
| | 80 | 79.3 ± 4.5 | 98.8 |

[a]In the presence of 0.1 mM RhB at pH 6.
[b]Confidence interval at n = 5 and P = 0.95.

**Table 3. Comparison of the developed DI-SDME-OP method with methods reported in the literature for Rh6G determination combining solid-phase or liquid-phase microextraction with spectrophotometric detection.**

| Method of microextraction[a] | Samples | Organic solvent / sample volume | Extraction time, min | Detection limit (nM) | Linear range (nM) | Reference |
|---|---|---|---|---|---|---|
| MSA-DLLME-HPLC | water, soft drink, cosmetic product | 500 μL of acetone, 1.05 mL of n-octanol/50 mL | 6 | 2.5 | 16–2090 | 7 |
| DLLME | wastewater | 3 mL of acetone, 300 μL of chloroform / 10 mL | 5 | 5 | 10–1900 | 9 |
| DLLME | food | 2.5 mL of ethanol, 250 μL of chloroform / 8 mL | 3 | 1 | 10–1000 | 10 |
| PM-D-μ-SPME | wastewater | 4 mL of methanol / 20 mL | 5 | 100 | 100–14000 | 11 |
| Magnetic MIP-SPE-HPLC | wine, paprika, river water | 9 mL of methanol / 10 mL | 30 | 2.7 | 20–100000 | 14 |
| REC-SBME | soft drink, lipstick, water | 25 μL of 1-octanol / 50 mL | 10 | 1.9 | 5–2500 | 19 |
| DI-SDME-OP | waters, lipstick | 55 μL of amyl acetate/7.5 mL | 30 | 3.4 | 10–500 | This work |

[a]MSA–magnetic stirring assisted; REC-SBME–rotating extraction cell solvent bar microextraction (mode of HF-LPME); PM-D-μ-SPME–polypyrrole-magnetite dispersive micro-solid-phase microextraction; MIP–molecularly imprinted polymer.

ratio of the sample and extract phases limited the sensitivity that could be achieved by the pre-concentration method. After the extraction was completed, centrifugation was used. Careful operations were necessary to separate the phases and transfer the extract to a microcuvette, making the whole procedure more time consuming and difficult to handle. The method of Biparva et al. required evaporation of the sedimented phase and its reconstitution with methanol [9]. When solvents lighter than water were used, the use of a home-made glass extraction chamber was necessary [7].

A small volume of organic solvent was used in the method of Badiie et al. based on the HF-LPME preconcentration technique [19]. This method, as well as our proposed method, can be employed without pretreatment to analyze complex samples. In addition, both methods use small amounts of reagents and organic solvents. Therefore, they can be considered as environmentally friendly and safe methods. Nevertheless, the HF-LPME method utilizes a more complicated procedure and a homemade device. The equilibrium condition cannot be achieved in this type of microextraction due to the constant loss of solvent from the pores of the hollow fiber, resulting in poor reproducibility.

Methods using solid-phase extraction for preconcentration and spectrophotometry for detection or HPLC with a spectrophotometric detector have sensitivities that do not exceed the sensitivity of our developed method [7, 14]. The gain in sensitivity obtained in these methods in the first step by using larger sample volumes and shorter extraction times is lost to a greater extent when a significant volume of eluent is used in the final step of the analysis. Only the use of a fluorescence or a mass spectrometric detector can significantly increase the sensitivity [13, 18]. However, when analyzing real objects, target signal measurements at low fluorescent dye concentrations are strongly influenced by reflected or scattered light as well as fluorescence from other species.

The use of OP and direct immersion microextraction contributes to the goals of "green" analytical chemistry. Thus, the number of sample preparation steps of the analytical procedure is reduced, since pre-concentration, phase separation, concentrate transfer to the instrument and on-line measurement of the analytical signal are integrated into one step with increased safety for the operator and reduced waste. The environmental friendliness of the proposed method was compared with methods combining liquid-phase microextraction and

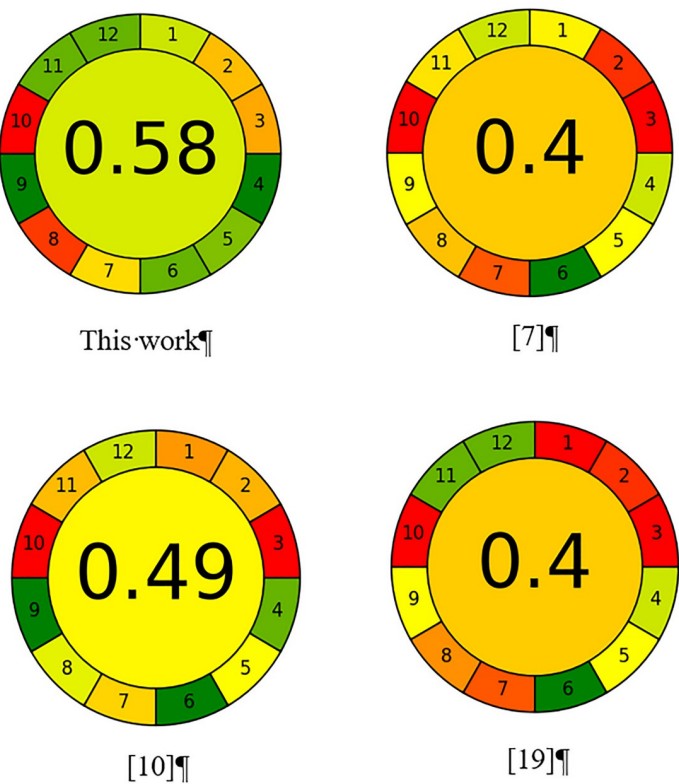

**Fig 6. The greenness score for procedures used for Rh6G determination combining microextraction and spectrophotometry: Combination of an optical probe and single-drop direct immersion microextraction (this work), magnetic stirring assisted dispersive liquid–liquid microextraction [7], derivative ratio spectrophotometry after dispersive liquid–liquid microextraction [10], hollow fiber liquid-phase microextraction [19].**

quantification by HPLC or SP methods using the Analytical Greenness (AGREE) metric [25], as shown in Fig 6 (See S1 Table in S1 File). The developed method obtained the best score (0.58) in terms of greenness when compared with previously published methods. Methods [7, 19] consumed a much larger sample volume, which also resulted in more waste. More sample preparation steps were used in the above methods. The power consumption of HPLC is higher than that of UV-visible spectrometry. Large volumes of toxic chloroform were used in procedures described in [7, 9].

## 4. Conclusion

The number of papers dealing with the coupling of spectrophotometry and liquid-phase microextraction remains relatively small. The difficulties associated with measuring absorbance in a small volume of the extraction phase prevent the coupling of single-drop liquid-phase microextraction methods with UV-Vis detection. Placing the extraction phase in the hole of an optical immersion probe enables the pre-concentration, phase separation, transfer of the extraction phase to the instrument and online measurement to be combined in one step. The DI-SDME-OP method was first applied for the preconcentration and quantification of organic dyes in waters and other types of samples. Comparison with existing methods combining separation and detection showed that the sensitivity of the proposed approach is not lower than that of existing spectrophotometric microextraction methods or methods combining

solid-phase extraction with UV-Vis spectroscopy or HPLC-UV-Vis. It should be noted that the proposed method has great potential for improvement, since only about 13% of Rh6G is extracted during the first 30 min. Consequently, the sensitivity can be significantly improved either by increasing the efficiency of the microextraction process or by increasing the extraction time.

The proposed DI-SDME-OP method is simple and has good selectivity and reproducibility. The proposed method uses only 55 μL of organic solvent to preconcentrate Rh6G, which profitably distinguishes it from other similar hybrid methods. The proposed method scores (0.57 points) the highest among previous methods according to the AGREE metric. The downside of the approach is the slow mass transfer at low analyte concentrations, though the use of the fixed-time kinetics method partially addresses this drawback. The search for solvents that have both good extraction ability and adhesion to the material of the optical probe, as well as ways to accelerate the mass transfer of the analyte, can significantly expand the application of optical probes for preconcentration by the direct immersion mode of SDME.

## Supporting information

**S1 File. AGREE assessment of the procedures used for Rh6G determination combining microextraction and spectrophotometry.**
(DOCX)

## Author Contributions

**Conceptualization:** Andriy Vishnikin.

**Data curation:** Arina Skok, Andriy Vishnikin, Yaroslav Bazel.

**Formal analysis:** Andriy Vishnikin, Ján Toth.

**Investigation:** Arina Skok, Andriy Vishnikin.

**Methodology:** Arina Skok, Andriy Vishnikin.

**Project administration:** Andriy Vishnikin, Yaroslav Bazel.

**Resources:** Andriy Vishnikin, Yaroslav Bazel.

**Software:** Ján Toth.

**Supervision:** Andriy Vishnikin, Yaroslav Bazel.

**Validation:** Arina Skok, Ján Toth.

**Writing – original draft:** Arina Skok, Andriy Vishnikin.

**Writing – review & editing:** Andriy Vishnikin, Yaroslav Bazel.

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
