## [Decision Letter · Decision Letter 0]

27 May 2024

PONE-D-24-18021Determination of Rhodamine 6G with direct immersion single drop microextraction combined with an optical probePLOS ONE

Dear Dr. Vishnikin,

Thank you for submitting your manuscript to PLOS ONE. After careful consideration, we feel that it has merit but does not fully meet PLOS ONE’s publication criteria as it currently stands. Therefore, we invite you to submit a revised version of the manuscript that addresses the points raised during the review process.

We look forward to receiving your revised manuscript.

Kind regards,

Thanh-Danh Nguyen, PhD

Academic Editor

PLOS ONE

Journal Requirements:

Reviewers' comments:

Reviewer's Responses to Questions

**Comments to the Author**

1. Is the manuscript technically sound, and do the data support the conclusions?

Reviewer #1: Partly

Reviewer #2: Yes

Reviewer #3: Yes

2. Has the statistical analysis been performed appropriately and rigorously? 

Reviewer #1: No

Reviewer #2: Yes

Reviewer #3: N/A

3. Have the authors made all data underlying the findings in their manuscript fully available?

Reviewer #1: Yes

Reviewer #2: Yes

Reviewer #3: Yes

4. Is the manuscript presented in an intelligible fashion and written in standard English?

Reviewer #1: No

Reviewer #2: Yes

Reviewer #3: No

5. Review Comments to the Author

Reviewer #1: The manuscript in question is on the development of a single-drop microextraction (SDME) approach combined with the use of an optical probe to determine a single analyte, Rhodamine 6G, by spectrophotometry.

The authors previously published a very similar work in Talanta ((2024) 269, 125511) (reference 18) in which the same analyte is considered, as well as identical reagents (picric acid and amyl acetate) in the SDME process. The only difference is in the detection mode (laser- and light emitting diode-based fluorescence previously).

Here's the abstract from the Talanta publication:

The use of an optical probe for fluorescence detection combined with direct immersion single-drop microextraction has been demonstrated as an innovative approach. The optical probe served both as a drop holder for extractant and as a measuring device which made it possible to eliminate the use of cuvettes. A laser and a light emitting diode (LED) were tested as possible light sources. Both of them showed comparable results. However, given the much smaller half-band width of the laser radiation, its use has proven to be preferable since background correction can be eliminated. Direct immersion single-drop microextraction of an ionic association complex of rhodamine 6G with picric acid with subsequent fluorescent detection (λex was 532 nm and 525 nm for laser and LED, respectively; λem was 560 nm for both laser and LED) was used a model system to evaluate the new approach. The extractant phase was a 55 μL amyl acetate microdrop fixed in the optical part of the probe. LOD, LOQ and linear calibration range were found as 0.14, 0.48 and 0.5–10 nmol L−1, and 0.15, 0.50 and 0.5–5 nmol L−1 for laser and LED light sources, respectively. The accuracy of the method was assessed by analyzing real water samples.

It is felt that there is nothing novel in the latest work. It appears to be a repeat of the Talanta study, with just a different detection system.

If an effort is made to expand the applicability to more analytes and differentiate from what was done before, it would make this more interesting.

Reviewer #2: Authors proposed a manuscript involving the determination of Rhodamine 6G with direct immersion single drop microextraction combined with an optical probe.

In general, the manuscript is well written and properly conducted. This proposal is interesting since an microextraction-based approach was succesfully developed, optimized, and applied in real samples.

However, some points need to be revised before the accepance in Plos One according to the comments below:

- The chemical structure and some physico-chemical properties of the analyte are important to be included at least in Supplementary Material.

- A trend in sample preparation techniques is the use of metrics for the evaluation of the sustainable aspects of analytical methodologies Some recent strategies such as AGREEprep and sample preparation metrics of sustainability (SPMS) have been employed to quantify how sustainable a methodology is. It should be interesting to include one of these metrics in order to permit a more comprehensive evaluation of the environmental aspects of this methodology.

- Authors have mentioned that “RSD values for different concentrations of Rh6G were ranged from 1.7 to 6.0 %”. Is intraday or interday precision? Which concentrations were examined? Please, include this information in the manuscript. Both precisions (intraday and interday) are important to be inserted.

- Sample preparation time should be included in Table 3 for the comparison of analysis throughput.

Reviewer #3: PONE-D-24-18021

This paper is an interesting and potentially useful application for on-site sampling and determination of dye chemicals in aqueous systems, as well as laboratory analyses of cosmetic products, using low-cost and portable UV-VIS probes and instrumentation. The authors should consider addition some relevant experimental data. In addition, while the manuscript is generally well written, there are some grammatical issues, such as sentence structure and word choices that need to be addressed.

Missing data:

1. A few sentences should be added describing the procedure (chemicals, time, etc.) required for cleaning the probe during field use

2. Line 203: what is meant by “using a smaller stirrer”. Is this the magnetic stirrer or stir be. If the stir be, be specic for size and type.

3. Line 98: picric acid is ususlly used in a moist state, to avoid detonation. Describe the material and source.

4. Units: the use of units changes from µg/L to µM. If both are to be used, when doing so the other should be included in perentheses.

5. Line 333: what is meant by a data curation? Please use a more meaningful description.

6. Are the natural waters (river) filtered before analysis?

7. The authors emphasize the greenness of this procedure versus other methods. This would be be much more effectively demonstrated by adding a paragraph with comparative calculations of their procedure to 1 or 2 of the other procedures discussed using AGREEprep. Just the results can be mentioned, more detailed methodology can be placed in the electronic addendum, if desired.

Grammatical issues

1. A number of sentences are too brief. These could be rewritten or combined to make a more flowing document.

2. Line 62: “such methods” is too vague”, rephrase.

3. Line 63: replace “processes” with “steps”.

4. Lines 73-78: In addition to the references, you should indicate that your research team did this, and the present work is a continuation of the previous work.

5. Lines 111-112: consider rearranging the sentence structure (buffer, pH) and use” stir bar” (consider point 2 above).

6. Line 116: Again consider rewriting.

7. Line 230: consider adding “reproducible”.

8. Line 238: Please rewrite (significant already???)

9. Line 240: “has been shown” – where, when, reference?

10. Line 267: consider rephrasing (such a toxic)

11. Line 317: consider rephrasing (not inferior not only)

6. PLOS authors have the option to publish the peer review history of their article (what does this mean?). If published, this will include your full peer review and any attached files.

Reviewer #1: No

Reviewer #2: No

Reviewer #3: No

---

## [Author Response · Author response to Decision Letter 0]

5 Jul 2024

Responses to Reviewer's questions

The manuscript has been corrected to meet PLOS ONE`s style requirements.

The statement has been added to Section 2.2.

A complete Data Availability Statement has been provided in the submission form.

Comments to the Author

5. Review Comments to the Author

Reviewer #1: The manuscript in question is on the development of a single-drop microextraction (SDME) approach combined with the use of an optical probe to determine a single analyte, Rhodamine 6G, by spectrophotometry.

The authors previously published a very similar work in Talanta ((2024) 269, 125511) (reference 18) in which the same analyte is considered, as well as identical reagents (picric acid and amyl acetate) in the SDME process. The only difference is in the detection mode (laser- and light emitting diode-based fluorescence previously).

Here's the abstract from the Talanta publication:

The use of an optical probe for fluorescence detection combined with direct immersion single-drop microextraction has been demonstrated as an innovative approach. The optical probe served both as a drop holder for extractant and as a measuring device which made it possible to eliminate the use of cuvettes. A laser and a light emitting diode (LED) were tested as possible light sources. Both of them showed comparable results. However, given the much smaller half-band width of the laser radiation, its use has proven to be preferable since background correction can be eliminated. Direct immersion single-drop microextraction of an ionic association complex of rhodamine 6G with picric acid with subsequent fluorescent detection (λex was 532 nm and 525 nm for laser and LED, respectively; λem was 560 nm for both laser and LED) was used a model system to evaluate the new approach. The extractant phase was a 55 μL amyl acetate microdrop fixed in the optical part of the probe. LOD, LOQ and linear calibration range were found as 0.14, 0.48 and 0.5–10 nmol L−1, and 0.15, 0.50 and 0.5–5 nmol L−1 for laser and LED light sources, respectively. The accuracy of the method was assessed by analyzing real water samples.

It is felt that there is nothing novel in the latest work. It appears to be a repeat of the Talanta study, with just a different detection system.

If an effort is made to expand the applicability to more analytes and differentiate from what was done before, it would make this more interesting.

Indeed, there is a certain similarity between this manuscript and a previously published article in Talanta 2024; 269: 125511. The main ideas of this manuscript are a new, original, much simpler and more efficient method for the quantitative determination of dyes, and a more detailed investigation of a number of questions of analytical importance. We believe that the proposed manuscript differs significantly from the published article in Talanta in the following criteria:

1. In the manuscript, we propose to use a much simpler detection system: instead of using a laser and then measuring the fluorescence intensity of the microextracts, the analyte is detected by UV-Vis. This method is much more convenient, simpler and can be used in conventional analytical laboratories without any problems. The use of rhodamine 6G as an analyte was due to the need to have a test system to evaluate the effectiveness of the new method, which would be more difficult to do with another dye.

2. The method of combining laser with microextraction determination of analytes, first described by us in the above-mentioned Talanta article, is quite effective, but has a number of disadvantages, such as the complexity of setting up the measurement process itself, the need to match laser parameters and analyte properties, compliance with special criteria for accurately setting the laser position relative to the extraction drop, etc. This can be a significant obstacle to the use of the proposed approach in laboratory and production environments. The use of a much simpler and easily installed optical probe-based detection has its advantages, in addition to the possibility of reproducing such a system in conventional laboratories, such as its versatility, suitability for the determination of many analytes, and quite satisfactory analytical properties, such as sensitivity, accuracy, and precision criteria.

3. The manuscript pays special attention to the selectivity of the determination. In particular, the possibility of determining rhodamine 6G in the presence of other dyes similar in structure and properties, such as fluorescein, eriochrome T and rhodamine B, is shown.

4. Another important difference is the expansion of the range of real samples in which rhodamine 6G can be determined. The effectiveness of this method for the analysis of not only water samples but also cosmetic products has been shown.

5. In contrast to the previous article, the “greenness” of the method by AGREE metric was evaluated. It is shown that the method with photometric detection is significantly superior in terms of a set of parameters to previously proposed methods with preliminary sorption or extraction preconcentration and detection by liquid chromatography, as it does not require the use of large volumes of toxic chloroform, has lower energy consumption and volume of sample and waste.

Reviewer #2: Authors proposed a manuscript involving the determination of Rhodamine 6G with direct immersion single drop microextraction combined with an optical probe.

In general, the manuscript is well written and properly conducted. This proposal is interesting since an microextraction-based approach was succesfully developed, optimized, and applied in real samples.

However, some points need to be revised before the accepance in Plos One according to the comments below:

- The chemical structure and some physico-chemical properties of the analyte are important to be included at least in Supplementary Material.

The chemical structure and mentioning the high solubility of Rhodamine 6G in water have been added into the text (Fig 1, Line 42).

- A trend in sample preparation techniques is the use of metrics for the evaluation of the sustainable aspects of analytical methodologies Some recent strategies such as AGREEprep and sample preparation metrics of sustainability (SPMS) have been employed to quantify how sustainable a methodology is. It should be interesting to include one of these metrics in order to permit a more comprehensive evaluation of the environmental aspects of this methodology.

Analytical Greenness (AGREE) metric was used to compare the greenness of the proposed method with three other existing methods using similar sample preparation technique (solid-phase and liquid-phase microextraction) (section 3.8).

- Authors have mentioned that “RSD values for different concentrations of Rh6G were ranged from 1.7 to 6.0 %”. Is intraday or interday precision? Which concentrations were examined? Please, include this information in the manuscript. Both precisions (intraday and interday) are important to be inserted.

Evaluation of interday and intraday precision for the proposed method was added to the section 3.6. 

- Sample preparation time should be included in Table 3 for the comparison of analysis throughput.

The extraction times for each method considered have been added to Table 3.

Reviewer #3: PONE-D-24-18021

This paper is an interesting and potentially useful application for on-site sampling and determination of dye chemicals in aqueous systems, as well as laboratory analyses of cosmetic products, using low-cost and portable UV-VIS probes and instrumentation. The authors should consider addition some relevant experimental data. In addition, while the manuscript is generally well written, there are some grammatical issues, such as sentence structure and word choices that need to be addressed.

Missing data:

1. A few sentences should be added describing the procedure (chemicals, time, etc.) required for cleaning the probe during field use

Operations required for sample preparation of river water and cleaning of the optical probe between measurements were added to section 2.2.

2. Line 203: what is meant by “using a smaller stirrer”. Is this the magnetic stirrer or stir be. If the stir be, be specific for size and type.

The dimensions of the magnetic stirring bar have been added.

3. Line 98: picric acid is ususlly used in a moist state, to avoid detonation. Describe the material and source.

An indication of the qualification of the picric acid preparation and the method of preparing the solution has been added to section 2.1.

4. Units: the use of units changes from µg/L to µM. If both are to be used, when doing so the other should be included in perentheses.

The same units for the Rh6G concentration expressed in µg L–1 were used throughout the article. 

5. Line 333: what is meant by a data curation? Please use a more meaningful description.

The term “data curation” was recommended by the rules of the journal. However, it was changed to a more meaningful “data evaluation”.

6. Are the natural waters (river) filtered before analysis?

Yes, river water was previously filtered. Information was added to the section 2.2.

7. The authors emphasize the greenness of this procedure versus other methods. This would be be much more effectively demonstrated by adding a paragraph with comparative calculations of their procedure to 1 or 2 of the other procedures discussed using AGREEprep. Just the results can be mentioned, more detailed methodology can be placed in the electronic addendum, if desired.

Analytical Greenness (AGREE) metric was used to compare the greenness of the proposed method with three other existing methods using similar sample preparation technique (solid-phase and liquid-phase microextraction).

Grammatical issues

1. A number of sentences are too brief. These could be rewritten or combined to make a more flowing document.

Several sentences have been merged or rewritten.

2. Line 62: “such methods” is too vague”, rephrase.

The related sentence is rewritten. Lines 83-86 in a new version of Ms.

3. Line 63: replace “processes” with “steps”.

Corrected. 

4. Lines 73-78: In addition to the references, you should indicate that your research team did this, and the present work is a continuation of the previous work.

We add an indication that our research team conducted the study described in reference [18] (lines 76, 77), as well as a mention that this paper is a continuation of a previous paper (line 117).

5. Lines 111-112: consider rearranging the sentence structure (buffer, pH) and use” stir bar” (consider point 2 above).

The sentence has been rewritten (Lines 149-151).

6. Line 116: Again consider rewriting.

The sentence has been rewritten (Lines 155-157).

7. Line 230: consider adding “reproducible”.

The word “accurate” has been replaced by “reproducible” (Line 276).

8. Line 238: Please rewrite (significant already???)

The word “significant ” has been replaced by “observed” (Line 297).

9. Line 240: “has been shown” – where, when, reference?

The sentence has been rewritten (Lines 298-300).

10. Line 267: consider rephrasing (such a toxic)

The sentence has been rephrased (Lines 326-328).

11. Line 317: consider rephrasing (not inferior not only)

The sentence has been rephrased (Lines 395-399).

In addition, the English language of the manuscript was checked by a native English speaker.

---

## [Decision Letter · Decision Letter 1]

23 Jul 2024

PONE-D-24-18021R1Determination of Rhodamine 6G with direct immersion single-drop microextraction combined with an optical probePLOS ONE

Dear Dr. Vishnikin,

Thank you for submitting your manuscript to PLOS ONE. After careful consideration, we feel that it has merit but does not fully meet PLOS ONE’s publication criteria as it currently stands. Therefore, we invite you to submit a revised version of the manuscript that addresses the points raised during the review process.

We look forward to receiving your revised manuscript.

Kind regards,

Thanh-Danh Nguyen, PhD

Academic Editor

PLOS ONE

Journal Requirements:

Additional Editor Comments:

It should be minor revision

Reviewers' comments:

Reviewer's Responses to Questions

**Comments to the Author**

1. If the authors have adequately addressed your comments raised in a previous round of review and you feel that this manuscript is now acceptable for publication, you may indicate that here to bypass the “Comments to the Author” section, enter your conflict of interest statement in the “Confidential to Editor” section, and submit your "Accept" recommendation.

Reviewer #3: (No Response)

2. Is the manuscript technically sound, and do the data support the conclusions?

Reviewer #3: Yes

3. Has the statistical analysis been performed appropriately and rigorously? 

Reviewer #3: N/A

4. Have the authors made all data underlying the findings in their manuscript fully available?

Reviewer #3: Yes

5. Is the manuscript presented in an intelligible fashion and written in standard English?

Reviewer #3: Yes

6. Review Comments to the Author

Reviewer #3: PONE-D-24-18021R1

The authors have addressed most of the points of the reviewers of the original manuscript. However, a few important issues remain as well as a couple of suggested minor changes.

Minor issues:

1. Lines 160-161: This sentence has no context? Something left out?

2. Line 203: Change to “salting out”

Major issues:

1. In Table 1, the units are mM, in Table 2, nM, in Table 3 µg L-1. For readers who do not wish to look up the molar or atomic mass of something in order to convert between units, this is troublesome and irritating. It also makes it difficult to recognize the significance of 10 mM of carbonate in river water, compared to the linear range of 5-240 µg L-1 for your method without conversion to identical units.

2. In Table 3 it would be useful for comparison of methods to know, without having to go to the original publications, the volume of sample and final analytical instrumentation technique, to again accurately compare the significant differences in methodology and greenness.

3. For Figure 6, there should be a listing of the 4 methods in the footnote to identify each AGREE assessment.

4. For Figure 6, it would be helpful to the reader and context of the figure, to provide the calculations for each AGREE assessment. This would be best handled in the electronic supplemental, referenced in the Table and manuscript of course..

7. PLOS authors have the option to publish the peer review history of their article (what does this mean?). If published, this will include your full peer review and any attached files.

Reviewer #3: No

---

## [Author Response · Author response to Decision Letter 1]

31 Jul 2024

Responses to Reviewer's questions

Minor issues:

1. Lines 160-161: This sentence has no context? Something left out?

A previous review indicated that the manuscript needed to be corrected to meet PLOS ONE`s style requirements. The following was suggested. 2. In your Methods section, please provide additional information regarding the permits you obtained for the work. Please ensure you have included the full name of the authority that approved the field site access and, if no permits were required, a brief statement explaining why. The text of the manuscript has therefore been finalized to reflect this suggestion.

2. Line 203: Change to “salting out”

The text has been corrected in line with this comment.

Major issues:

1. In Table 1, the units are mM, in Table 2, nM, in Table 3 µg L-1. For readers who do not wish to look up the molar or atomic mass of something in order to convert between units, this is troublesome and irritating. It also makes it difficult to recognize the significance of 10 mM of carbonate in river water, compared to the linear range of 5-240 µg L-1 for your method without conversion to identical units.

Concentration units in the article (Table 3, abstract and Introduction) were converted to one unit system derived from mol L–1 (nM or mM).

2. In Table 3 it would be useful for comparison of methods to know, without having to go to the original publications, the volume of sample and final analytical instrumentation technique, to again accurately compare the significant differences in methodology and greenness.

The volume of sample has been added to Table 3. The caption to Table 3 indicates the final analytical instrumental methodology. In all compared methods, the analytical signal was measured by spectrophotometry.

3. For Figure 6, there should be a listing of the 4 methods in the footnote to identify each AGREE assessment.

4 methods have been listed in the caption to Fig. 6.

4. For Figure 6, it would be helpful to the reader and context of the figure, to provide the calculations for each AGREE assessment. This would be best handled in the electronic supplemental, referenced in the Table and manuscript of course.

S1 Table has been added and posted as Supporting Information. It contains the data needed to assess the AGREE score for the developed and three compared methods. The AGREE scores have been refined.

---

## [Editor Report · Decision Letter 2]

5 Aug 2024

Determination of Rhodamine 6G with direct immersion single-drop microextraction combined with an optical probe

PONE-D-24-18021R2

Dear Dr. Vishnikin,

We’re pleased to inform you that your manuscript has been judged scientifically suitable for publication and will be formally accepted for publication once it meets all outstanding technical requirements.

Kind regards,

Thanh-Danh Nguyen, PhD

Academic Editor

PLOS ONE

Additional Editor Comments (optional):

It can be accepted in the present form.
---

## [Editor Report · Acceptance letter]

9 Aug 2024

PONE-D-24-18021R2 

PLOS ONE

Dear Dr. Vishnikin, 

I'm pleased to inform you that your manuscript has been deemed suitable for publication in PLOS ONE. Congratulations! Your manuscript is now being handed over to our production team.

Kind regards, 

on behalf of

Dr. Thanh-Danh Nguyen 

Academic Editor

PLOS ONE